# Integrated Metabolomics and Transcriptomics Analyses Identify Key Amino Acid Metabolic Mechanisms in *Lacticaseibacillus paracasei* SMN-LBK

**DOI:** 10.3390/foods14050730

**Published:** 2025-02-21

**Authors:** Jie Shen, Yuyu Du, Yuechenfei Shen, Ning Kang, Zhexin Fan, Zhifeng Fang, Bo Yang, Jiancheng Wang, Baokun Li

**Affiliations:** 1Key Laboratory of Agricultural Product Processing and Quality Control of Specialty (Co-Construction by Ministry and Province), School of Food Science and Technology, Shihezi University, Shihezi 832000, China; shenjie@stu.shzu.edu.cn (J.S.); 20232011006@stu.shzu.edu.cn (Y.D.); shenyuechenfei@stu.shzu.edu.cn (Y.S.); 20222111016@stu.shzu.edu.cn (N.K.); zhexin_fan@shzu.edu.cn (Z.F.); zhifengf@jiangnan.edu.cn (Z.F.); 2Key Laboratory for Food Nutrition and Safety Control of Xinjiang Production and Construction Corps, School of Food Science and Technology, Shihezi University, Shihezi 832000, China; 3Engineering Research Center of Storage and Processing of Xinjiang Characteristic Fruits and Vegetables, Ministry of Education, School of Food Science and Technology, Shihezi University, Shihezi 832000, China; 4Functional Food Center, Key Laboratory of Xinjiang Medicinal Plant Resources Utilization, Ministry of Education, Shihezi 832000, China; 5Engineering Research Center of Camel Milk of Xinjiang, Ili 835100, China; 6State Key Laboratory of Food Science and Resources, School of Food Science and Technology, Jiangnan University, Wuxi 214122, China; bo.yang@jiangnan.edu.cn; 7International Joint Research Laboratory for Maternal-Infant Microbiota and Health, Jiangnan University, Wuxi 214122, China

**Keywords:** *Lacticaseibacillus paracasei* SMN-LBK, glutamate, arginine, mechanisms

## Abstract

During lactobacillus fermentation, the types of proteins in the fermentation substrate significantly influence the characteristics of the fermented product. Proteins are composed of various amino acids. Consequently, investigating the metabolic mechanisms of key amino acids during lactic acid bacteria fermentation is important for improving their application in the food industry. In this study, the growth of *Lacticaseibacillus paracasei* SMN-LBK was significantly inhibited following glutamate and arginine deficiency (*p* < 0.05). Genomic analysis and in vitro addition assays showed that α-ketoglutarate (OXO), as a precursor of glutamate, significantly eliminated growth inhibition of SMN-LBK (*p* < 0.05). Meanwhile, the inhibition of SMN-LBK growth following arginine deficiency may be linked to glutamate. Metabolomics analysis illustrated that glutamate and arginine deficiencies mainly affected the carbohydrate and amino acid metabolic pathways of SMN-LBK, especially the pentose phosphate pathway, alanine, glutamate and aspartate metabolism, and arginine metabolism. Transcriptomics analysis further identified glutamate and arginine deficiencies affecting carbohydrate and amino acid metabolism, specifically the glutamate metabolism, pentose phosphate pathway, and glycolysis/gluconeogenesis, involving key genes such as pfkA, gapA, ldh, argG, argE, and argH. Elucidating the molecular mechanisms of key amino acids in SMN-LBK will provide a theoretical foundation for understanding the differential fermentation of various proteins by lactic acid bacteria.

## 1. Introduction

Lactic acid bacteria and their metabolites have been extensively employed across a broad spectrum of applications in the food, pharmaceutical, and medical sectors, especially certain *Lacticaseibacillus paracasei* (*L. paracasei*) [1,2,3]. *L. paracasei* is a Gram-positive bacterium [4], which is found across a variety of ecological niches, including the gastrointestinal and genitourinary tracts, dairy products, fermented vegetables, and silage [5,6]. *L. paracasei* inhibits pathogenic bacteria, regulates the immune system, and possesses antioxidant properties that promote the balance of microflora and produce various enzymes [7,8,9,10]. Currently, *L. paracasei*, like most lactic acid bacteria, is mainly used in fermented dairy products due to its capability to metabolize carbohydrates and amino acids [11,12,13,14,15].

SMN-LBK, used in this experiment, is a novel probiotic strain. Previous studies have found that SMN-LBK has good fermentation and probiotic properties, and it encoded numerous proteases and peptidases in genetic sequence, which contributed to a more comprehensive protein hydrolysis system [16,17]. In a previous study, we examined the impact of single-strain fermentation (SMN-LBK) and mixed-strain fermentation (SMN-LBK and *Kluyveromyces marxianus* SMN-S7-LBK) on cheese flavor, and found that a number of potentially functional flavor compounds are involved in amino acid metabolism, with the glutamate and glutamine metabolic pathways playing a crucial role in flavor formation [18]. Meanwhile, the presence of glutamate and arginine was found to be crucial for the formation of antioxidant peptides during the fermentation of bi-protein (walnut milk) by SMN-LBK [13]. Through these findings, it was evident that the type of amino acid is a significant factor influencing the differential changes in SMN-LBK under various fermentation conditions. Therefore, investigating the molecular metabolic mechanisms of key amino acids could provide valuable insights into the phenomenon of differences in the production of proteins from different sources by SMN-LBK fermentation.

Currently, the application of multi-omics technologies in elucidating the molecular mechanisms of lactic acid bacteria is increasingly prevalent. Wang investigated the secondary metabolite changes and functional active substance biotransformation mechanisms in *Bifidobacterium animalis* subsp. lactis HN-3 fermented *Elaeagnus angustifolia* var. *orientalis* (L.) Kuntze juice (EOJ) in secondary metabolite changes and mechanisms of biotransformation of functionally active substances. However, single metabolomic analysis resulted in the study that did not deeply investigate the relationship between enzymes and metabolites inherent in molecular metabolic mechanisms [19]. Therefore, the joint application of multi-omics will be more beneficial for clarifying the molecular metabolic mechanisms in organisms. For instance, transcriptomics and metabolomics analyses were used to investigate the alterations in gene expression and metabolite profiles in *Lactobacillus rhamnosus* Probio-M9 and its spaceflight-induced mutant R7970, thereby providing a comprehensive and systematic elucidation of the molecular mechanisms underlying the phenotypic variations observed in the spaceflight-induced mutant R7970 [20]. Similarly, by using combined genomics and metabolomics analyses, the phenylalanine metabolic pathway, the amino acid metabolic catabolic and anabolic pathways, and the fatty acid biosynthetic pathway were found to be the major biosynthetic pathways involved in the formation of fermentation flavor in lactic acid bacteria [21]. Combined multi-omics analyses have been successfully employed by a growing number of researchers to reveal key molecular metabolic mechanisms in lactic acid bacteria in practical applications. Consequently, this study aimed to elucidate the pivotal mechanisms of amino acid metabolism during the growth of SMN-LBK by integrating genomic, transcriptomic, and metabolomic analyses.

In this study, the limiting amino acids in the growth of SMN-LBK were identified through the single amino acid omission technique. α-Ketoglutarate, the precursor of glutamate was predicted by genomic information, could restore the growth of SMN-LBK. The fact that SMN-LBK was significantly inhibited after arginine deficiency was supported by the genomic information that aspartate and glutamate may be precursors of arginine. Metabolomics and transcriptomics analyses were used to explore the key metabolites and genes in SMN-LBK after the deletions of glutamate and arginine, and the results revealed that both differential metabolites and genes were enriched in amino acid metabolism and carbohydrate metabolism. The findings from this research will offer a theoretical and scientific foundation for understanding the differential metabolism of SMN-LBK during fermentation with animal and plant proteins.

## 2. Materials and Methods

### 2.1. Strain

*Lacticaseibacillus paracasei* SMN-LBK, isolated from horse milk wine in Tacheng, Xinjiang, was deposited in the Chinese Typical Cultures Depository Center (CCTCC NO: M 2017429).

### 2.2. Strain Activation, Culture and Sample Collection

SMN-LBK was incubated in MRS liquid medium for 18 h, and then cultured on solid MRS medium for 48 h. Individual colonies were picked and cultured in CDM liquid medium for 18 h to obtain activated strains [1]. MRS solid medium: MRS medium (Qingdao, China), 2% agar (Beijing, China). CDM medium: thiamine hydrochloride, riboflavin, niacin, calcium pantothenate, lipoic acid, and cobalamin, 0.001 g/L; pyridoxine hydrochloride, 0.002 g/L; biotin, 0.0025 g/L; folic acid, orotate, pyridoxamine nicotinate, inosine, and thymidine, 0.005 g/L; ascorbic acid and glutamate, 0.5 g/L; threonine, 0.225 g/L; tryptophan and calcium chloride anhydrous, 0.05 g/L; tyrosine, 0.25 g/L; valine, 0.325 g/L; cysteine, 0.13 g/L; alanine, 0.24 g/L; arginine and methionine, 0.125 g/L; aspartate, 0.42 g/L; glycine, 0.175 g/L; histidine, 0.15 g/L; isoleucine 0.21 g/L; leucine, 0.475 g/L; lysine, 0.44 g/L; phenylalanine, 0.275 g/L; proline, 0.675 g/L; serine, 0.34 g/L; p-aminobenzoic acid, adenine, guanine, uracil, and xanthine, 0.01 g/L; anhydrous sodium acetate, 8.3 g/L; dextrose, 20 g/L; dipotassium hydrogen phosphate and potassium dihydrogen phosphate, 3 g/L; diammonium hydrogen citric acid, 1 g/L; manganese chloride tetrahydrate, 0.025 g/L; magnesium sulfate heptahydrate, 0.2 g/L [22,23]. Vitamins, amino acid and nucleotide reagents were obtained from Shanghai Macklin Biochemical Technology Corporation (Shanghai, China).

The bacterial suspensions were inoculated into CDM medium, glutamate-deficient CDM medium and arginine-deficient CDM medium for 24 h. Bacterial suspensions from logarithmic (10 h) and stable (18 h) growth periods were taken, and the supernatant was discarded to collect the organisms. Collected organisms should be rapidly frozen in liquid nitrogen for subsequent transcriptomics and metabolomics analyses.

### 2.3. Determination of Growth Curves

The activated SMN-LBK was washed 2 to 3 times with saline, and inoculated into CDM liquid medium with a single amino acid deficiency determining the OD600 value every 2 h. SMN-LBK grew in complete CDM medium as the control [24,25].

### 2.4. Prediction of Amino Acid Metabolism Pathway

KEGG analysis of the SMN-LBK genome predicted the metabolic pathways for each amino acid. A unified mechanism pathway was integrated with simplified the metabolic pathway of each amino acid. Furthermore, the interactions between amino acids and the roles of precursor substances were elucidated, providing a theoretical basis for the subsequent results of individual amino acid deficiency assays [24]. SMN-LBK genomic data are available at NCBI by searching for accession number NZ_CP101831.1.

### 2.5. Addition of α-Ketoglutarate

SMN-LBK was inoculated into CDM medium for glutamate deficiency with α-ketoglutarate concentrations of 0.25 g/L, 0.5 g/L, 0.75 g/L, 1 g/L, and 5 g/L, each at a 4% (*v*/*v*). CDM mediums with and without glutamate were served as control groups [26]. The OD600 values were determined for 18 h.

### 2.6. Metabolomics Analysis

300 μL 80% methanol solution (*v*/*v*) was added to bacterial sample; the mixture was vortexed for 30 s and sonicated for 6 min. Subsequently, the samples were centrifuged at 5000 rpm (1 min, 4 °C), and the supernatant was transferred to a new centrifuge tube and lyophilized into powder. Finally, the sample was dissolved in a 10% methanol solution, corresponding to the volume taken, and analyzed by LC-MS [27,28]. The operation of this experiment was performed on ice. The metabolome operation of LC-MS was conducted by Novozymes.

The obtained metabolomics data were transformed using metaX and analyzed using principal component analysis (PCA) [29]. The criteria for screening differential metabolites were VIP > 1, *p* < 0.05, and fold change > 1.5. Volcano plots and KEGG network diagrams were generated using the R (3.4.3) package ggplot2. Heatmap was performed using the OmicStudio tools.

### 2.7. Transcriptomics of SMN-LBK

RNA of SMN-LBK was extracted using the RNA prep Pure Cultured Cells/Bacteria Total RNA Extraction Kit (TIANGEN Biochemistry, Beijing, China). The extracted RNA underwent stringent quality control using an Agilent 2100 bioanalyzer (Agilent Technologie, Santa Clara, CA, USA). mRNA was enriched by removing rRNA with probes and randomly fragmented using a fragmentation buffer. Subsequently, a strand-specific library was constructed, following the method described by Parkhomchuk [30]. Illumina sequencing was performed on the qualified library. After sequencing, the filtered sequences were analyzed for genomic localization using Bowtie2 software (v2.3.4.3) [31]. The gene sequencing and annotation were conducted by Novozymes (Beijing, China).

In the processing of transcriptomics data, DESeq2 (1.20.0) was primarily utilized for differential expression analysis between the two groups of genes. Genes with a *p* < 0.05 and |log2FoldChange| > 0 were considered differentially expressed. The clusterProfiler (v3.8.1) software was utilized to conduct KEGG pathway analysis.

### 2.8. Statistical Analysis

Growth graphs were plotted using GraphPad Prism (8.0.2), and significance was analyzed using SPSS software (27.0.1). Data were presented as mean ± SEM. The significance analysis was performed using the Duncan test. Different letters indicated a significant difference at α = 0.05. *p* < 0.05 was considered statistically significant. The diagram of amino acid metabolism was drawn by Figdraw 2.0.

## 3. Results

### 3.1. Identification of Key Amino Acids in the Growth Process of SMN-LBK

The single amino acid omission technique was employed to investigate the impact of amino acids on the growth of SMN-LBK. It was observed that SMN-LBK reached the stabilization phase at 18 h when cultured in CDM medium (Figure 1A). Notably, most of the amino acids did not have a significant effect on SMN-LBK growth. However, the growth of SMN-LBK was nearly stagnant with a single deficiency of glutamate. And, a single deficiency of arginine significantly inhibited SMN-LBK growth, resulting in growth concentrations that were only 40% of those observed under CDM culture conditions at 18 h (Figure 1A). The results suggested that arginine and glutamate were growth-limiting amino acids for the growth of SMN-LBK (*p* < 0.05).

### 3.2. The Metabolic Pathway of Arginine and Glutamate in SMN-LBK

Genomic information predicted that α-ketoglutarate was a precursor to glutamate, and both glutamate and aspartate served as precursors for arginine synthesis. To confirm the role of α-ketoglutarate on the growth of SMN-LBK after glutamate deficiency, α-ketoglutarate addition experiments were performed in vitro. Addition of α-ketoglutarate could restore the growth of SMN-LBK. The restorative effect of α-ketoglutarate on glutamate deficiency increased with increasing concentrations, reaching a maximum of 78% of the CDM group when the concentration of α-ketoglutarate reached 1 g/L. However, 5 g/L α-ketoglutarate significantly inhibited SMN-LBK growth compared to 1 g/L (Figure 1C).

Arginine deficiency incompletely inhibited the growth of SMN-LBK, probably due to the fact that glutamate and aspartate could synthesize part of arginine, as revealed by the genomic information. This result further confirmed that glutamate was a growth-limiting amino acid for SMN-LBK (Figure 1B).

### 3.3. Non-Targeted Metabolomics Analysis

Non-targeted metabolomics studies were conducted to investigate the mechanisms of action of glutamate and arginine in SMN-LBK. A total of 1074 positive-ion mode metabolites and 427 negative-ion mode metabolites were identified. The top five classes of metabolites in both positive and negative ion modes included lipids and lipid-like molecules, organic acids and derivatives, nucleosides, nucleotides and analogs, benzenoids, and organoheterocyclic compounds (Appendix A). After merging positive and negative ions, the six replicates in the metabolomic data for each group were loosely but generally clustered together and differed significantly between groups (Figure 2A,B). Meanwhile, we found that amino acid metabolism, nucleotide metabolism, metabolism of cofactors and vitamins, carbohydrate metabolism and lipid metabolism were the main pathways of metabolites enrichment (Appendix A).

101, 127, 129, and 152 differential metabolites were obtained in the four comparative pairs (G1 VS C1, G2 VS C2, J1 VS C1, and J2 VS C2), respectively (Figure 2C–F). In particular, the metabolites N-acetyl-asp-glu and L-tryptophan were significantly up-regulated after glutamate deficiency, while phenylglyoxylic acid and D-fructose 6-phosphate were significantly down-regulated (Figure 2C,D). Arginine deficiency significantly up-regulated the level of 11-dehydro thromboxane B2, 2′-deoxycytidine 5′-diphosphate (dCDP), and 4-nitrophenol, nevertheless significantly down-regulated folic acid and adenosine (Figure 2E,F). The results showed that the absence of glutamate and arginine significantly affected the metabolites during the growth of SMN-LBK.

Additionally, KEGG network enrichment analysis was performed on the differential metabolites identified using the diffusion algorithm to reveal the effect of critical pathways on SMN-LBK growth (*Lacticaseibacillus paracasei* N1115 as reference strain). This analysis revealed that the metabolites were predominantly associated with pathways of carbohydrate metabolism and amino acid metabolism (Figure 3). Enrichment of amino acid synthesis, cysteine and methionine metabolism, and thiamine metabolism occurred only after glutamate deficiency (Figure 3A,B). In contrast, enrichment of the TCA cycle, glycine, serine, and threonine metabolism, tyrosine metabolism, vancomycin resistance nicotinate and nicotinamide metabolism, protein export, and bacterial secretion system were only observed in arginine deficiency (Figure 3C,D). While enrichment of the pentose phosphate pathway, purine metabolism, pyrimidine metabolism, alanine, aspartate and glutamate metabolism, phenylalanine metabolism, tryptophan metabolism, nucleotide metabolism, and the ABC transporter pathway were present in all four comparative pairs (Figure 3A–D). It was noteworthy that the enrichment of DNA replication, base excision repair, nucleotide excision repair, and mismatch repair pathways may result from the accumulation of organic acids in SMN-LBK during the stable phase (Figure 3B,D) and the enrichment of these pathways is closely related to growth. Meanwhile, arginine synthesis and arginine metabolism occurred only in the logarithmic phase of the growth process in glutamate-deficient and arginine-deficient groups (Figure 3A,C). This may be related to the fact that arginine utilization was mainly present in the logarithmic phase of SMN-LBK growth.

### 3.4. Transcriptomics Analysis

Transcriptomics analysis was utilized to investigate the impact of glutamate and arginine deficiencies on genes expression in SMN-LBK. The principal component analysis of the transcriptomics data revealed the samples from each group clustered together well and the differences between groups were significant, which was similar to the clustering observed in the metabolomics data (Figure 4A). Concurrently, the transcriptomics analysis indicated that the deletions of glutamate and arginine significantly influenced the expression levels of the associated genes (Figure 4B and Appendix A).

The KEGG enrichment analysis of the identified differential genes, which were detected after glutamate and arginine deficiencies, revealed that carbohydrate metabolism and amino acid metabolism were the primary pathways involved, which was consistent with the findings from differential metabolites enrichment in metabolomics. Notably, the ribosome pathway, which is central to the RNA-to-protein process in bacteria and plays a crucial role in bacterial growth, was only significantly enriched in the glutamate deficiency group (Figure 4C–F). At the same time, the glycolysis/gluconeogenesis, pyruvate metabolic pathways, alanine, aspartate and glutamate metabolism, presenting in glutamate and arginine deficiencies groups, were closely linked to the key metabolic pathways identified through metabolomics analysis.

### 3.5. Joint Metabolomics and Transcriptomics Analyses

Based on the metabolome and transcriptome enrichment results, the differential genes in the pentose phosphate pathway, purine metabolism, pyrimidine metabolism, and the metabolism of alanine, aspartate, and glutamate, as well as in the glycolysis/gluconeogenesis and pyruvate metabolic pathways, were specifically quantified (Appendix A). Differential genes with a high percentage in these enrichment pathways were an important factor contributing to the differences between glutamate and arginine deficiencies in SMN-LBK growth. Therefore, clustering analyses of differentially key genes and metabolites associated with these pathways revealed that changes in these key genes and metabolites were strongly correlated with amino acid species and culture periods (Figure 5). Notably, the expression of genes controlling the energy production process (gpmA, pgi, fba, pfkA, pykF, and ldh) was higher in the single glutamate-deficient group (Group G) than in the CDM group (Group C) and the single arginine-deficient group (Group J). Meanwhile, group C had more ADP, which can be used to synthesize ATP in organisms.

Additionally, the metabolic machinery containing glutamate and arginine was mapped by joint analysis of metabolic pathways obtained from differential metabolites and differential genes enrichment (Figure 6G). This mechanism comprised the pentose phosphate pathway, purine metabolism, pyrimidine metabolism, alanine, aspartate, and glutamate metabolism pathways, glycolysis/gluconeogenesis, and the arginine metabolism (Figure 6G). In the above pathway, the glycolysis/gluconeogenesis pathway and the TCA cycle, which was derived from the metabolism of alanine, glutamate, and aspartate, were the energy supply pathways in this mechanism.

The amount of citric acid in the TCA cycle was significantly higher in both the arginine-deficient and glutamate-deficient groups than the CDM groups (Figure 6A). This result suggested that the glycolysis and TCA pathways associated with citrate synthesis were abnormal following glutamate and arginine deficiencies. The genes argG and argH involved in arginine metabolic pathway were barely expressed in the glutamate-deficient and CDM groups (Figure 6D,F), and the expression of argE was lower in the glutamate-deficient group relative to the other two groups (Figure 6E). The expression of the genes argG, argH, and argE can promote arginine synthesis. In addition, ornithine and citrulline in the arginine metabolic pathway were significantly higher in the logarithmic phase in the arginine-deficient group than in the glutamate-deficient and CDM groups (Figure 6B,C). This result suggested that the presence of more genes and metabolites involved in arginine metabolism was an important method for SMN-LBK to cope with the absence of arginine in the environment.

## 4. Discussion

This study investigated the effects of various amino acids on the growth of SMN-LBK. Glutamate, alanine, cysteine, histidine, leucine, and isoleucine are essential amino acids for the growth of most lactic acid bacteria [32]. However, the results in our study indicated that arginine and glutamate were essential for the growth of SMN-LBK, primarily influencing carbohydrate metabolism and amino acid metabolism. It has been suggested that arginine and glutamate were crucial for the adaptation of LAB to acidic environments, as their deaminated metabolites served as key energy substrates for cell growth and metabolism [33]. Glutamate and arginine were abundant in certain plant proteins, with walnuts being a particularly rich source. These amino acids were pivotal for the generation of desirable flavors and the synthesis of antioxidant peptides during the fermentation of dairy products [13,34]. Consequently, a comprehensive and systematic investigation into the molecular mechanisms of glutamate and arginine in SMN-LBK holds substantial significance.

Owing to the extensive application of multi-omics analytical methodologies, it has become increasingly feasible to achieve a comprehensive and systematic elucidation of the alterations in microbial growth mechanisms in response to external environmental influences. In this study, metabolomics and transcriptomics analyses revealed significant differences in metabolites and genes among the glutamate-deficient group, arginine-deficient group, and CDM group, which were mainly enriched in carbohydrate metabolism and amino acid metabolism. Previous studies also found that altering the environment in which lactobacilli live can affect bacterial metabolism, particularly concerning amino acid and carbohydrate metabolism [20,35,36]. Carbohydrate metabolic pathways were intricately linked to various amino acid metabolic pathways, playing crucial roles in energy provision and the bacteria’s ability to withstand environmental stresses during growth [20].

By analyzing KEGG metabolic network maps, it was found that the TCA cycle was more functional in the arginine-deficient group compared to the glutamate-deficient group. This may be related to the high expression of fumAB and genes of the arginine synthesis pathway (argG, argE, argH, and argE) in the arginine deficiency group. The TCA cycle was the primary pathway for sugar metabolism in the growth of lactic acid bacteria, alongside the glycolysis pathway [37]. Previous genomic analyses revealed that SMN-LBK lacked a complete set of TCA cycle enzymes, preventing it from metabolizing CO2 as eukaryotes do [16]. There was a close association between the TCA cycle and glutamate. Incompleteness of the TCA pathway was the main cause of citric acid accumulation in glutamate deficiency group. Glutamate was crucial for the normal growth of bacteria, serving as a precursor to α-ketoglutarate in the TCA cycle. Concurrently, α-ketoglutarate can restore normal growth in SMN-LBK when it acted as a precursor to glutamate. In addition, it was noteworthy that the α-ketoglutarate in this mechanism was not directly involved in the TCA cycle, but rather, served as a precursor to glutamate in the arginine synthesis pathway, which produced fumarate that was involved in the TCA cycle. However, excessive addition of α-ketoglutarate (5 g/L) in vitro limited the growth of SMN-LBK. It was also reported that more than 6 mM α-ketoglutarate inhibited glutamate production, thereby negatively influencing the reaction of the AT enzyme of *Lb. paracasei* subsp. *paracasei* LC01 [38]. Therefore, it was concluded that the incomplete TCA cycle in SMN-LBK stopped energy supply after glutamate deficiency. Moreover, the presence of glutamate facilitated the glutamate decarboxylase system to play an important role in pH regulation and thus resisted acid stress [39]. Meanwhile, differences in growth between the glutamate-deficient and arginine-deficient groups were observed, which can also be attributed to the presence of the TCA cycle. In addition, several metabolic pathways were differentially enriched and may contribute to the growth differences observed in SMN-LBK between glutamate-deficient and arginine-deficient groups. These important metabolic pathways included amino acid synthesis, cysteine and methionine metabolism, thiamine metabolism, glycine, serine, and threonine metabolism, tyrosine metabolism, nicotinate and nicotinamide metabolism, protein export, and bacterial secretion system.

Based on the differential metabolites in the K-Means analysis (Appendix A), it was hypothesized that the metabolic mechanisms of glutamate and arginine were similar in SMN-LBK. This mechanism involved the arginine synthesis and metabolism. In this pathway, nine genes controlled eight enzymes, encoding for five regulators. The expression of these genes was regulated by the product of the argR gene [40,41]. There were three pathways for L-arginine synthesis in microorganisms: (1) The linear pathway, where L-glutamate was first catalyzed by argA to form N-acetylglutamate, which was then further catalyzed by various enzymes encoded by argB to argH to synthesize arginine [42,43]. (2) The cyclic pathway, in which N-acetylornithine was catalyzed by argJ to transfer the acetyl group to L-glutamate, forming N-acetylglutamate, and subsequently arginine was synthesized through the action of individual enzymes encoded by argB to argH [44]. (3) Acetyl ornithine and carbamoyl phosphate were first catalyzed by argF’ to form acetyl citrulline, then by argE to form L-citrulline, and finally with aspartate to synthesize arginine under the catalytic action of argG and argH [45]. However, in this study, the genes argJ and argF’ were not expressed in SMN-LBK through transcriptomics and metabolomics analyses. Citrulline was generated via carbamoyl phosphate and ornithine was catalyzed by ornithine carbamoyl transferase. Meanwhile, key metabolites in the arginine synthesis and metabolism pathway, including glutamate, glutamine, acetylornithine, ornithine, citrulline, aspartate, and fumaric acid, were identified, along with the genes argG, argH, OTC, carB, argE, and argR (Figure 5A,B). This finding supported the hypothesis that arginine was involved in the first synthetic pathway in SMN-LBK. In the analysis of the arginine metabolic pathway, urea was not detected in the metabolomics data, and genes encoding arginases were not identified in the transcriptomics data. However, given the presence of ornithine transcarbamoylase, it was hypothesized that arginine metabolism in SMN-LBK may proceed via the arginine deiminase pathway (ADI pathway). Some studies have found that lactic acid bacteria can increase intracellular pH and metabolic activity of bacteria through the ADI pathway [33,46]. Therefore, in this study, the absence of arginine may impair the functionality of the ADI pathway, potentially rendering SMN-LBK incapable of withstanding the acidic stress of the culture environment, thereby halting its growth.

The combined use of multi-omics techniques has been shown to be an effective method for studying bacterial metabolic mechanisms. In this study, joint metabolomics and transcriptomics analyses were used and identified. The glycolysis/gluconeogenesis pathway was the main energy supply pathway during the growth of SMN-LBK. A suite of enzymes facilitated the conversion of glucose to pyruvate through glycolysis/gluconeogenesis pathway, which subsequently underwent either reduction to lactate or entry into the tricarboxylic acid (TCA) cycle. Notably, fructose-6-phosphate kinase (pfkA) was the rate-limiting enzyme in the initial steps of glycolysis, glyceraldehyde-3-phosphate dehydrogenase (gapA) was crucial for the phosphorylation and oxidation of each triose phosphate molecule, and L-lactate dehydrogenase (ldh) played a key role in converting pyruvate to lactate [37,47]. Therefore, the differential expression of key enzymes in glycolysis/gluconeogenesis also attributed to the differences in the growth of SMN-LBK which was observed between glutamate deficiency and arginine deficiency. Highly expressed genes of the glycolysis/gluconeogenesis pathway (gpmA, pgi, fba, pfkA, pykF, and ldh) in the single glutamate-deficient group were the response of SMN-LBK to maintain normal functioning in vivo after glutamate deficiency. However, the presence of glutamate was necessary for the growth of SMN-LBK, so the high expression of the relevant genes did not restore the growth of SMN-LBK. The pentose phosphate pathway, a crucial component of glucose oxidative catabolism, did not primarily generate ATP but served to produce essential substances, such as NADPH, ribose 5-phosphate, and PRPP, that were required for SMN-LBK.

## 5. Conclusions

In summary, the absence of glutamate and arginine significantly inhibited the growth of SMN-LBK. Metabolomics and transcriptomics analyses revealed that the growth of the bacterium after glutamate and arginine deficiencies was primarily related to carbohydrate metabolism and amino acid metabolism. Meanwhile, key metabolites, including citric acid, ornithine, and citrulline, as well as the expression of key genes such as pfkA, gapA, ldh, argG, argE, and argH, were strongly associated with the growth of SMN-LBK. Ultimately, limited energy supply and the essential functions in glutamate and arginine were the key factors that impaired SMN-LBK growth. This study provided valuable theoretical insights for SMN-LBK better fermenting varied origin proteins and offering a new research direction for optimizing the nitrogen source in fermentation processes.

## Figures and Tables

**Figure 1 foods-14-00730-f001:**
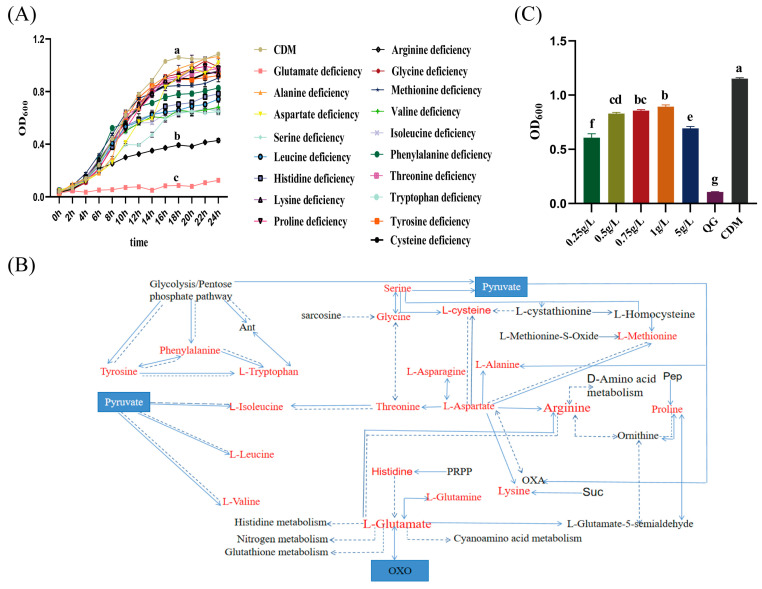
Identification of key amino acids during SMN-LBK growth. (**A**) Growth curves of SMN-LBK with single amino acids deficiencies. (**B**) Metabolic mechanism diagram for 22 amino acids, with red markers indicating the amino acids required for this experiment. Single arrows in the diagram show the unidirectional flow of metabolic pathways, bi-directional arrows indicate reversible reactions, and dotted lines suggest possible metabolic circulations. (**C**) Effects of α-ketoglutarate on the growth of SMN-LBK. QG stands for glutamate deficiency group. In Figure (**C**), the QG and CDM groups serve as controls, whereas the groups supplemented with α-ketoglutarate are designated as experimental groups. Significance analyses were performed using the Duncan test. Different letters indicate a significant difference at α = 0.05.

**Figure 2 foods-14-00730-f002:**
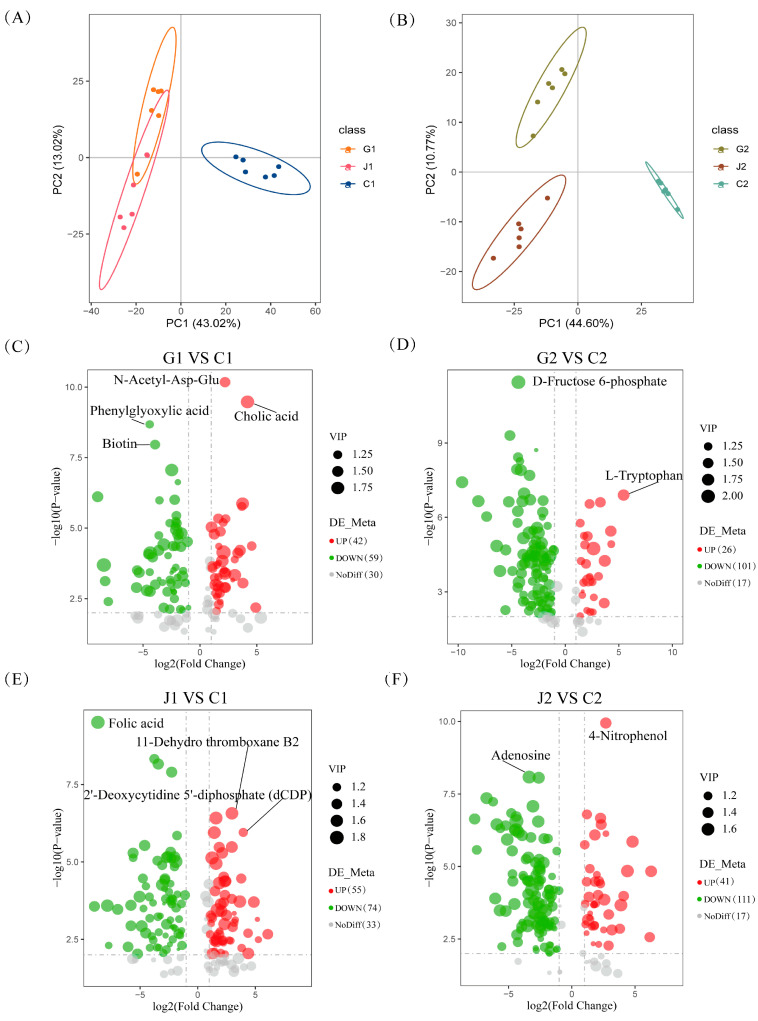
Analysis of metabolites in SMN-LBK after glutamate and arginine deficiencies. (**A**) Principal component analysis score plot during the logarithmic growth period. (**B**) Principal component analysis score plot during the stable growth period. (**C**–**F**) Volcano diagrams of the four comparative pairs. In this study, experimental groups G and J consisted of SMN-LBK cultured in glutamate-deficient and arginine-deficient CDM medium, respectively. Control groups C consisted of SMN-LBK cultured in CDM medium. The number 1 indicates a logarithmic growth period (10 h) and 2 indicates a stable growth period (18 h).

**Figure 3 foods-14-00730-f003:**
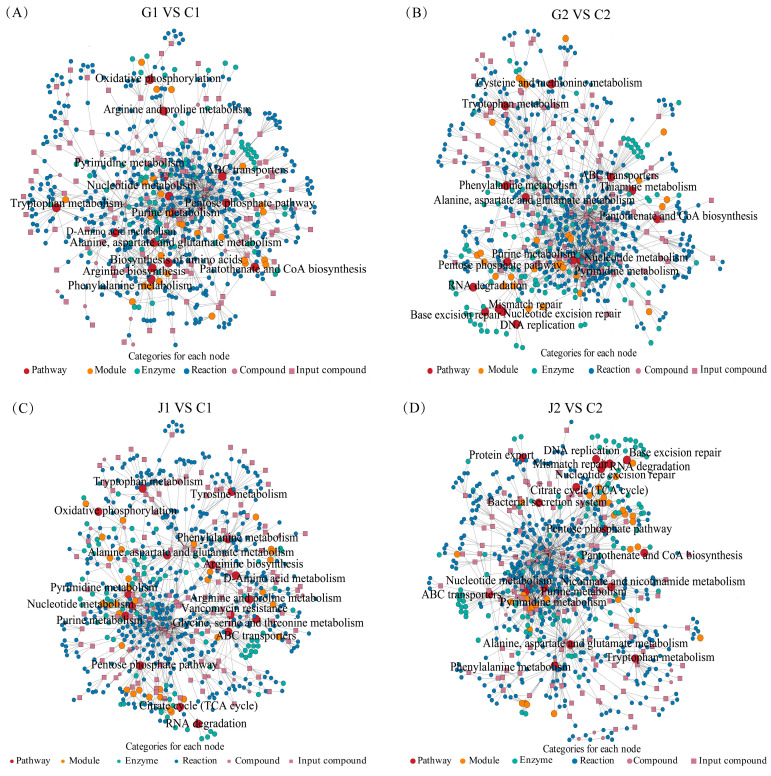
Enrichment of metabolic pathways. (**A**–**D**) KEGG metabolic network diagrams for four comparative pairs of groups. G, J, and C denote the glutamate-deficient group, arginine-deficient group, and CDM group, respectively. The number 1 indicates a logarithmic growth period (10 h) and 2 indicates a stable growth period (18 h). Groups G and J serve as the experimental groups, while Group C serves as the control group. The red circles marked in the figure represent metabolite pathways.

**Figure 4 foods-14-00730-f004:**
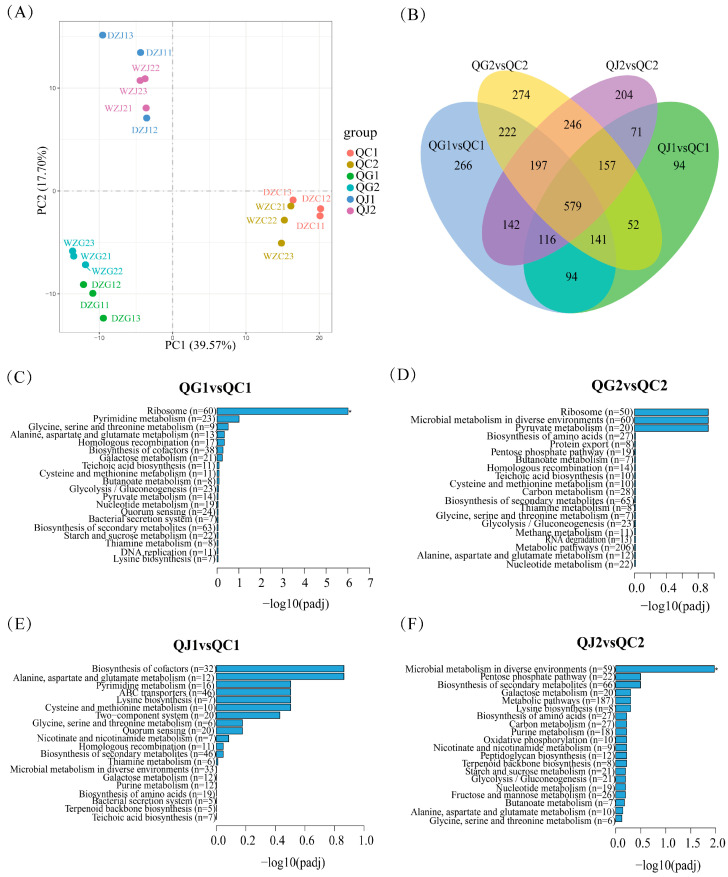
Analysis of genes in SMN-LBK after glutamate and arginine deficiencies. (**A**) Principal component analysis score plot. (**B**) Venn diagram showing the overlap of differential genes. (**C**–**F**) The KEGG enrichment results for four comparison pairs. In the transcriptomics analysis, QG, QJ, and QC denote the glutamate-deficient group, arginine-deficient group, and CDM group, respectively. In this experiment, QG and QJ represent the experimental groups, while QC represents the control group. Adjusted *p* value (Padj) indicates more significant enrichment pathways, and * indicates Padj < 0.05. The number 1 after the letter indicates a logarithmic growth period (10 h) and 2 indicates a stable growth period (18 h).

**Figure 5 foods-14-00730-f005:**
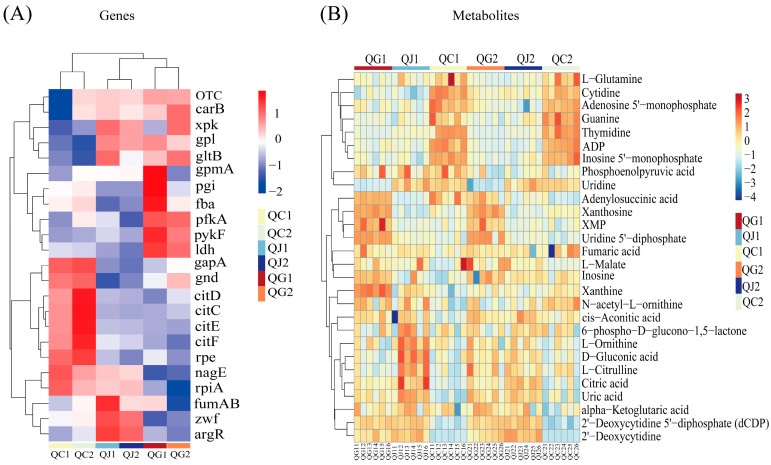
Analysis of key genes in important pathways. (**A**) Heatmap of genes. (**B**) Heatmap of metabolites. QG, QJ, and QC denote the glutamate-deficient group, arginine-deficient group, and CDM group, respectively. In this experiment, QG and QJ represent the experimental groups, while QC represents the control group. The number 1 after the letter indicates a logarithmic growth period (10 h) and 2 indicates a stable growth period (18 h).

**Figure 6 foods-14-00730-f006:**
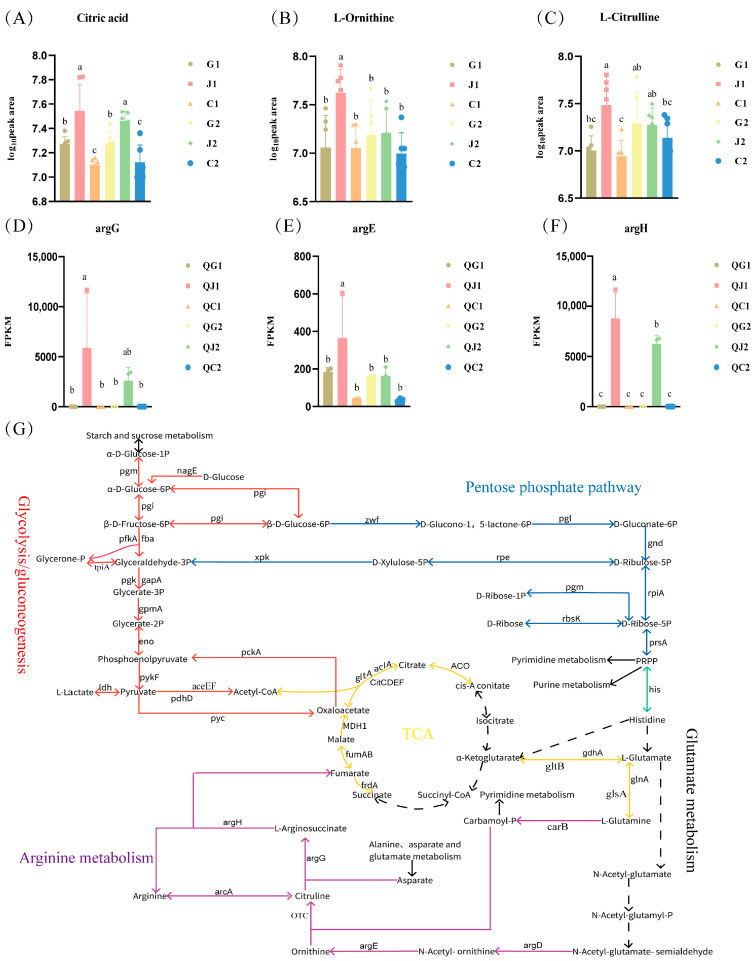
Integrated metabolomics and transcriptomics analyses. (**A**) Citric acid content. (**B**) Ornithine content. (**C**) Citrulline content. (**D**) argG expression. (**E**) argE expression. (**F**) argH expression. (**G**) Molecular mechanisms influencing SMN-LBK growth following glutamate and arginine deficiencies. The red line indicates the glycolysis/gluconeogenesis pathway. Blue, the pentose phosphate pathway; Yellow, the TCA cycle. Purple, the glutamate metabolism and arginine metabolism pathways. Green, histidine synthesis pathway. Solid black, connectivity to other pathways. Dashed black, a possible or a non-existent pathway. G and J in the metabolomic data represent the experimental group, whereas C represents the control group. QG and QJ in the transcriptomic data represent the experimental group and QC represents the control group. Significance analyses were performed using the Duncan test. Different letters indicate a significant difference at α = 0.05.

## Data Availability

The original contributions presented in this study are included in the article/Appendix A. Further inquiries can be directed to the corresponding authors.

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
