# Peer review of "Integrated Metabolomics and Transcriptomics Analyses Identify Key Amino Acid Metabolic Mechanisms in Lacticaseibacillus paracasei SMN-LBK"

_foods, 2025, doi:10.3390/foods14050730_

Round 1
Reviewer 1 Report
Comments and Suggestions for Authors
The paper titled "Integrated metabolomics and transcriptomics analyses identify key amino acid metabolic mechanisms in Lacticaseibacillus paracasei SMN-LBK" investigates the metabolic pathways of amino acids in the lactic acid bacterium Lacticaseibacillus paracasei SMN-LBK. The study employs a combination of metabolomics and transcriptomics to understand how deficiencies in glutamate and arginine affect the growth and metabolic processes of this bacterial strain,
It is necessary to present some chromatograms and a table of metabolites found.
Comments on the Quality of English LanguageNot comment.
Author Response
Dear reviewer,
Thank you very much for your professional suggestion concerning our manuscript entitled “Integrated metabolomics and transcriptomics analyses identify key amino acid metabolic mechanisms in Lacticaseibacillus paracasei SMN-LBK”. Based on your request, We have provided information on the metabolites found in this study. In addition, The changes marked in red in the revised manuscript.
Kind Regards
Sincerely yours,
Baokun Li
Comment: It is necessary to present some chromatograms and a table of metabolites found.
Response: Thank you for your positive feedback on our manuscript. Since metabolomic data are not uploaded on other platforms and may be needed in follow-up work, we can only provide partial information on metabolites. We place this result in “Non-published Material”.
Reviewer 2 Report
Comments and Suggestions for Authors
The manuscript by Jie et al. presents interesting findings supported by well-designed methodology and a thoughtful discussion of the results. However, there are areas that require significant improvement to enhance the clarity and scientific rigor of the study. I recommend a major revision to address these issues before the manuscript can be considered for publication.
-The abstract lacks detailed quantitative data to support its claims. For example, phrases like "significantly inhibited" and "closely linked" require numerical or statistical backing to provide context and strengthen conclusions Line 10-32. Add specific metrics (e.g., growth inhibition percentages, p-values) and emphasize the practical application of the findings.
-The introduction mentions "obvious differences" in SMN-LBK’s fermentation ability for animal and plant proteins but does not provide concrete examples or previous studies to justify this statement Line 56-64. Cite specific studies or include preliminary results to contextualize this observation.
- The connection between multi-omics approaches and their importance to SMN-LBK metabolism is mentioned but not explained in detail Line 86-90. Provide examples of how multi-omics has been successfully used in similar studies.
-The description of strain activation is overly detailed, focusing on medium components. This could be condensed for clarity and supplemented with references for the medium composition Line 122-125.
-The claim that arginine and glutamate are growth-limiting amino acids is made without direct reference to statistical tests or growth data Line 252-257. Provide p-values or fold-change data to substantiate these claims.
-The description of metabolomics results is unclear due to jargon-heavy language. Terms like "clear clustering pattern" need to be illustrated with specific examples or metrics Line330-343. Rephrase for clarity and simplify the discussion for a broader audience.
- Provide more detail on why the single amino acid omission technique was chosen. Was it validated against alternative methods?
- Explain why the specific concentrations of α-ketoglutarate (0.25–5 g/L) were chosen. Is this range physiologically relevant?
-The integration of metabolomics and transcriptomics lacks coherence. While pathways are mentioned, specific genes or metabolites critical to them are not discussed in sufficient depth.
-KEGG pathway analysis is central to the study but not well elaborated in terms of biological significance. Focus on pathways showing the strongest enrichment and explain their relevance to L. paracase.
-Provide more details on the experimental controls used, especially in the metabolomics and transcriptomics studies.
-Were biological replicates used for RNA sequencing and LC-MS analysis? Mentioning sample sizes and validation steps would enhance reproducibility.
-The manuscript touches on the role of the TCA cycle and glycolysis but does not explore why these pathways are incomplete in L. paracasei. Is this a strain-specific trait or common in lactic acid bacteria?
-Improve the resolution and labeling of all figures, particularly Figures 3 and 6.
-Include clear legends for the metabolic maps, indicating the roles of key genes and metabolites.
-Compare findings to previous studies on Lacticaseibacillus species. For instance, how do the results align with the metabolic traits of L. paracasei in dairy versus plant-based fermentations?
-Address discrepancies, such as why citric acid levels were elevated despite TCA cycle incompleteness.
Author Response
Dear reviewer,
Thank you very much for your critical comments and professional suggestions concerning our manuscript entitled “Integrated metabolomics and transcriptomics analyses identify key amino acid metabolic mechanisms in Lacticaseibacillus paracasei SMN-LBK”. Based on your suggestions and requests, we have made corrected modification on the revised manuscript and the detailed corrections are listed below. The changes marked in red in the revised manuscript.
Kind Regards
Sincerely yours,
Baokun Li
Comment 1: The abstract lacks detailed quantitative data to support its claims. For example, phrases like "significantly inhibited" and "closely linked" require numerical or statistical backing to provide context and strengthen conclusions Line 10-32. Add specific metrics (e.g., growth inhibition percentages, p-values) and emphasize the practical application of the findings.
Response 1: Thank you for your constructive comment. We have added the corresponding p-values in the abstract. However, specifically labeling the growth inhibition percentages in the abstract would have greatly exceeded the word count of the abstract as required by this journal, so we only put these data into the main text. In addition, for the phrase "closely linked", it was mainly a conclusion inferred from the maps of each amino acid pathway obtained from the genomics data in this study. We have revised this sentence for the sake of more rigorous academic expression (Lines 35-38).
Comment 2: The introduction mentions "obvious differences" in SMN-LBK’s fermentation ability for animal and plant proteins but does not provide concrete examples or previous studies to justify this statement Line 56-64. Cite specific studies or include preliminary results to contextualize this observation.
Response 2: Thank you for your comment. As the data reflecting these changes have not yet been published, we have revised the relevant sentence. These revisions do not impact the overall understanding of the manuscript (Lines 67-76).
Comment 3: The connection between multi-omics approaches and their importance to SMN-LBK metabolism is mentioned but not explained in detail Line 86-90. Provide examples of how multi-omics has been successfully used in similar studies.
Response 3: Thank you for your valuable suggestion. We have revised the relevant sentence (Lines 81-89, Lines 94-100).
Comment 4: The description of strain activation is overly detailed, focusing on medium components. This could be condensed for clarity and supplemented with references for the medium composition Line 122-125.
Response 4: Thank you for your valuable suggestion. We have grouped descriptions of substances with the same content. In addition, we have added relevant references (Lines 124-135).
Comment 5: The claim that arginine and glutamate are growth-limiting amino acids is made without direct reference to statistical tests or growth data Line 252-257. Provide p-values or fold-change data to substantiate these claims.
Response 5: Thank you for your comment. We have added the p-value at the appropriate place in the manuscript (Line 202). The contents of Lines 197-201 present data on the growth of SMN-LBK after glutamate and arginine deficiency. In addition, data on growth between the three groups (10h and 18h) were analysed by SPSS software to clarify the significance between the groups (below). Significance analysis are performed using the Duncan test. Different letters indicate a significant difference at α = 0.05.
Comment 6: The description of metabolomics results is unclear due to jargon-heavy language. Terms like "clear clustering pattern" need to be illustrated with specific examples or metrics Line330-343. Rephrase for clarity and simplify the discussion for a broader audience.
Response 6: Thank you for your valuable suggestion. We have revised the sentences where the description was unclear (Lines 233-235, Lines 288-290).
Comment 7: Provide more detail on why the single amino acid omission technique was chosen. Was it validated against alternative methods?
Response 7: Thank you for your comment. The lack of multiple amino acids in the environment at the same time will make the final experimental results unfavourable for analysis and comparison. However, the single amino acid omission technique can be better used to explore the key role of each amino acid in the growth process of SMN-LBK. In addition, single amino acid omission techniques have been used to explore key amino acids in Lactobacillus plantarum (DOI: 10.7506/spkx1002-6630-20200805-078).
Comment 8: Explain why the specific concentrations of α-ketoglutarate (0.25–5 g/L) were chosen. Is this range physiologically relevant?
Response 8: Thank you for your question. Firstly, α-ketoglutarate was found to have the potential to be interconvertible with glutamate during genomic analyses. Secondly, a α-ketoglutarate addition test was performed to see initially if α-ketoglutarate addition could restore SMN-LBK growth in a single glutamate-deficient CDM medium (the α-ketoglutarate concentration was first chosen to be 0.5 g/L because because we wanted to ensure that the α-ketoglutarate was at the same concentration of glutamate as that in the CDM medium). The results showed that α-ketoglutarate significantly restored the growth of SMN-LBK after glutamate deficiency. Finally, based on the growth curves and the results of this pre-experiment, we took 0.5 g/L as the standard and 0.25 as the gradient. The value of 5g/L was mainly taken to explore whether high concentration of α-ketoglutarate would inhibit the growth. The single glutamate deficiency group can be interpreted as the 0 g/L α-ketoglutarate group.
Comment 9: The integration of metabolomics and transcriptomics lacks coherence. While pathways are mentioned, specific genes or metabolites critical to them are not discussed in sufficient depth.
Response 9: Thank you for your comment. We have added analyses of metabolites and genes at the corresponding positions in the manuscript (Lines 320-329, Lines 394-397, Lines 406-409, and Lines 467-472 )
Comment 10: KEGG pathway analysis is central to the study but not well elaborated in terms of biological significance. Focus on pathways showing the strongest enrichment and explain their relevance to L. paracase.
Response 10: Thank you for your valuable suggestion. The analysis of KEGG metabolic pathways in this manuscript focuses on the pathways co-enriched in the four comparative pairs, and then analyses these pathways in depth to reveal the metabolic mechanisms of key amino acid molecules in the regulation of SMN-LBK. For the most highly enriched pathways, we may briefly mention them in the manuscript. This is because the highly enriched pathways may be very important for the growth of SMN-LBK, but may not be very useful for mapping the final molecular metabolic mechanism (Lines 262-278 and Lines 299-304). We hope you are satisfied with our answer.
Comment 11: Provide more details on the experimental controls used, especially in the metabolomics and transcriptomics studies.
Response 11: Thank you for your comment. We have refined the experimental and control groups that were not clearly labeled in the manuscript (Lines 223-225, Lines 253-256, Lines 283-284, Lines 308-313, Lines 333-335, and Lines 365-367).
Comment 12: Were biological replicates used for RNA sequencing and LC-MS analysis? Mentioning sample sizes and validation steps would enhance reproducibility.
Response 12: Thank you for your question. The different groups in the RNA sequencing in this experiment contained three biological replicates and the different groups in the LC-MS analysis contained six biological replicates. In this manuscript, information about the biological replicates of the metabolome and transcriptome can be known from Figures 2 and 4, so it is not described in detail in the experimental methodology. I hope you are satisfied with our answer.
Comment 13: The manuscript touches on the role of the TCA cycle and glycolysis but does not explore why these pathways are incomplete in L. paracasei. Is this a strain-specific trait or common in lactic acid bacteria?
Response 13: Thank you for your comment. Since we have explained the incompleteness of the TCA cycle in SMN-LBK by genomic analysis in previous studies (mainly because SMN-LBK itself lacks the enzymes in the TCA cycle), this manuscript does not go into detail to analyse the reasons for the incompleteness of the TCA cycle [1].
[1] Wang, J. et al. Comparative genomic analysis of Lacticaseibacillus paracasei SMN-LBK from koumiss. Frontiers in Microbiology 13 (2022). https://doi.org/10.3389/fmicb.2022.1042117
Comment 14: Improve the resolution and labeling of all figures, particularly Figures 3 and 6.
Response 14: Thank you for your valuable suggestion. We have adjusted the resolution of these images.
Comment 15: Include clear legends for the metabolic maps, indicating the roles of key genes and metabolites.
Response 15: Thank you for your valuable suggestion. We have added clear legends for the metabolic maps (Fig. 6G).
Comment 16: Compare findings to previous studies on Lacticaseibacillus species. For instance, how do the results align with the metabolic traits of L. paracasei in dairy versus plant-based fermentations?
Response 16:Thank you for your valuable suggestion. The aim of this manuscript is mainly to clarify the molecular mechanism of key amino acids in SMN-LBK and to provide a theoretical basis for subsequent fermentations of this strain using both plant-type and animal-type proteins. However, L. paracasei has been little studied in fermentation of bi-proteins (plant and animal). Therefore, the references related to this aspect of research in this manuscript mainly focus on previous studies. And as this part of the references does not facilitate a direct comparison of the results, they are placed in the introduction section (Lines 67-76).
Comment 17: Address discrepancies, such as why citric acid levels were elevated despite TCA cycle incompleteness.
Response 17: Thank you for your comment. The accumulation of citric acid in the experimental groups (G and J) was significantly higher than that in the control group (C). This difference was attributed to a more refined molecular metabolic mechanism in SMN-LBK cultured in CDM complete medium, which led to the inevitable utilization of citric acid by SMN-LBK. For groups G and J, the imperfect TCA pathway is the main reason for the significant accumulation of citric acid (Lines 402-403).

Round 2
Reviewer 1 Report
Comments and Suggestions for Authors
The work presents the metabolomics analysis illustrated that glutamate and arginine deficiencies mainly affected the carbohydrate and amino acid metabolic pathways of SMN-LBK, especially the pentose phosphate pathway, alanine, glutamate and aspartate metabolism, and arginine metabolism. Transcriptomics analysis further identified glutamate and arginine deficiencies affecting carbohydrate and amino acid metabolism, specifically the glutamate metabolism, pentose phosphate pathway, and glycolysis/gluconeogenesis, involving key genes such as pfkA, gapA, ldh, argG, argE, and argH.
All changes suggested to the authors have been made. The work has improved.
Reviewer 2 Report
Comments and Suggestions for Authors
It could be accepted in its current form.